



# The aerosol radiative effects of uncontrolled combustion of domestic waste

John K. Kodros[1], Rachel Cucinotta[2], David A. Ridley[3], Christine Wiedinmyer[4], Jeffrey R. Pierce[1]

[1]Department of Atmospheric Science, Colorado State University, Fort Collins, Colorado, USA
[2]University of North Carolina at Charlotte, Charlotte, North Carolina, USA
[3]Department of Civil & Environmental Engineering, Massachusetts Institute of Technology, Cambridge, Massachusetts, USA
[4]National Center for Atmospheric Research, Boulder, Colorado, USA

*Correspondence to*: J. K. Kodros (jkodros@atmos.colostate.edu)

**Abstract.** Open, uncontrolled combustion of domestic waste is a potentially significant source of aerosol; however, this aerosol source is not generally included in many global emissions inventories. To provide a first estimate of the aerosol radiative impacts from domestic-waste combustion, we incorporate the Wiedinmyer et al. (2014) emissions inventory into GEOS-Chem-TOMAS, a global chemical-transport model with online aerosol microphysics. We find domestic-waste combustion increases global-mean black carbon and organic aerosol concentrations by 8% and 6%, respectively, and by greater than 40% in some regions. Due to uncertainties regarding aerosol optical properties, we estimate the globally averaged aerosol direct radiative effect to range from -5 mW m$^{-2}$ to -20 mW m$^{-2}$; however, this range increases to -40 mW m$^{-2}$ to +4 mW m$^{-2}$ when we consider uncertainties in emission mass and size distribution. In some regions with significant waste combustion, such as India and China, the aerosol direct radiative effect may exceed -0.4 W m$^{-2}$. Similarly, we estimate a cloud-albedo aerosol indirect effect of -13 mW m$^{-2}$, with a range of -4 mW m$^{-2}$ to -49 mW m$^{-2}$ due to emission uncertainties. In the regions with significant waste combustion, the cloud-albedo aerosol indirect effect may exceed -0.4 W m$^{-2}$.

## 1 Introduction

Open, uncontrolled combustion of domestic waste occurs on a global scale, potentially emitting significant amounts of black carbon (BC) and organic aerosol (OA); however, this source is not commonly considered in modeling studies estimating aerosol radiative impacts (Christian et al., 2010; Mohr et al., 2008; Wiedinmyer et al., 2014). Uncontrolled combustion of domestic waste refers to the intentional disposal of trash in an open fire either at home or at a dump (Wiedinmyer et al., 2014). This practice is common in developing countries in both rural and urban areas where municipal waste collection is insufficient; however, open combustion of domestic waste also occurs in rural developed countries (Christian et al., 2010; Wiedinmyer et al., 2014). The proximity of waste-combustion emissions to residences has created concerns for both air



quality and climate. In a separate study, we estimate the global burden of mortalities due to chronic exposure of waste-combustion $PM_{2.5}$ (Kodros et al., 2016). Here we provide a first estimate of the aerosol radiative impacts of domestic-waste combustion.

Carbonaceous aerosol from combustion sources can impact climate in several ways; the best understood of these being the direct radiative effect (DRE) and cloud-albedo aerosol indirect effect (AIE) (Boucher et al., 2013). The DRE refers to scattering and absorption of incoming solar radiation by aerosols (Charlson et al., 1992). The magnitude and relative amount of scattering and absorption is dependent on particle composition and size, as well as how the particles are mixed together (Bond et al., 2006; Bond et al., 2013). BC has a strong absorbing component, while OA is either considered as completely scattering or having some absorbing component in the shorter visible and UV wavelengths (i.e. brown carbon) (Andreae and Gelencsér, 2006; Hammer et al., 2015; Wang et al., 2014), although these optical properties of organic aerosol are an open area of research. Additionally, scattering and absorption efficiencies of carbonaceous particles depend on the ratio of particle diameter to wavelength of radiation. For solar radiation, atmospheric aerosol typically have peak efficiencies between 100 nm to 1 μm, though this will vary with composition (Seinfeld and Pandis, 2012). Finally, absorption tends to increase in an internally mixed population (all particles in a size range contain an identical mixture of all chemical species) compared to an externally mixed population (different chemical species exist in separate particles) (Chung and Seinfeld, 2005; Jacobson, 2001).

The cloud-albedo AIE refers to altering the reflectivity of clouds by changing the cloud droplet number concentration (CDNC) (Twomey, 1974). The ability for OA and BC cloud condensation nuclei (CCN) to activate into cloud droplets is a strong function of particle size. In typical atmospheric conditions, OA and BC (when mixed with hydrophilic species) typically act as CCN when they have diameters larger than 40-100 nm (Petters and Kreidenweis, 2007). The number of particles in this size range is dependent on primary emissions as well as nucleation, condensation, and coagulation (Pierce and Adams, 2009). As these microphysical processes are often inter-connected, the AIE response to an emission source may be non-linear (e.g. additional primary emissions may reduce nucleation and condensation growth while increasing coagulation) (Bauer et al., 2010; Butt et al., 2016; D'Andrea et al., 2015). The AIE is also dependent on particle hygroscopicity (Petters and Kreidenweis, 2007). While combustion aerosol tend to have lower hygroscopicity (kappa for OA=~0.1 and BC= 0.0), mixing with inorganic species, such as sulfate and sea salt, and chemical aging of the organics to more-hygroscopic species allows for activation at smaller diameters.

Waste combustion is an often-overlooked emission source, especially in developing countries (Christian et al., 2010). Wiedinmyer et al. (2014) provided a new global inventory for the uncontrolled combustion of domestic waste, estimating global BC and OC emissions of 0.6 and 5.1 Tg yr[-1], respectively. This estimate is an order of magnitude greater than earlier estimates of emissions from uncontrolled waste combustion in Bond et al. (2004) (which focused on combustion-aerosol emissions in general, not just from waste combustion). Amount of annual waste burned was the largest difference between the inventories, where Bond et al. (2004) estimated a total of 33 Tg and Wiedinmyer et al. (2014) predicted 970 Tg. Further, Wiedinmyer et al. (2014) considered waste combustion in both rural and urban areas in developing and rural areas



in developed countries. Many global inventories, including Emissions Database for Global Atmospheric Research (EDGAR, Janssens-Maenhout et al. 2010), do not include estimates of uncontrolled waste-combustion emissions. In addition to carbonaceous emissions, Wiedinmyer et al. (2014) estimated 0.5 Tg yr$^{-1}$ of sulfur-dioxide ($SO_2$) and 1.1 Tg yr$^{-1}$ of ammonia ($NH_3$). These gas-phase species may increase nucleation and condensation rates, thereby increasing particle number and

mass through secondary aerosol formation.

There are several uncertainties related to emissions in the Wiedinmyer et al. (2014) inventory. First, the reported uncertainty in the Akagi et al. (2011) emission factors for BC and OC are 42% and 93%. This is in addition to uncertainties in the mass of waste generated (i.e estimates of fuel use). Second, the Wiedinmyer et al. (2014) inventory does not include information on emission size distribution. The emitted particle size distribution varies with fuel type and flaming condition,

and goes through rapid aging in the first few hours after emission on spatial scales not resolved in global models (Janhäll et al., 2010; Sakamoto et al., 2016). Finally, this inventory does not include information on the optical properties of organic aerosol. Absorptive OA (or brown carbon, BrC) has been observed in some wood and agricultural waste fuels, under certain flaming conditions (Chen and Bond, 2010; McMeeking et al., 2014; Saleh et al., 2013); however, we are not aware of studies determining these aerosol properties for domestic-waste combustion. In this manuscript, we explore uncertainties in total

emission mass and size distribution (affecting both DRE and AIE) as well as optical properties (affecting just DRE).

To our knowledge, this is the first paper to look at the global aerosol radiative impacts from uncontrolled combustion of domestic waste. In Section 2, we discuss the model set-up (2.1), methods for model evaluation (2.2), and calculation of radiative effects (2.3-2.4). In Section 3, we show comparisons to ground observations of model simulations with and without waste-combustion emissions. In Section 4, we discuss changes in modeled aerosol mass (4.1), size-

resolved aerosol number concentration (4.2), direct radiative effect (4.3), and cloud-albedo aerosol indirect effect (4.4). In Section 5, we share our conclusions.

## 2 Methods

### 2.1 Model overview

To calculate aerosol number, mass, and size distributions from domestic-waste combustion, we use the Goddard Earth

Observing System global chemical-transport model (GEOS-Chem) version 10.01 coupled with TwO Moment Aerosol Sectional (TOMAS) microphysics scheme (Adams and Seinfeld, 2002). We use GEOS-Chem with 47 vertical layers and a 4°x5° horizontal resolution. This version of TOMAS uses 15 size sections ranging from 3 nm to 10 μm with tracers for sulfate, sea salt, OA, BC, and dust (Lee and Adams, 2012; Y. Lee et al., 2013). GEOS-Chem is driven using reanalysis meteorology fields from GEOS5 (http://gmao.gsfc.nasa.gov). Since meteorology is prescribed, the addition of waste-

combustion emissions does not feedback into weather patterns. All simulations are for the year 2010, with 1 month of model spin-up that is not included in analysis.





We use a ternary nucleation scheme involving water, sulfuric acid, and ammonia following the parameterization of Napari et al. (2002), scaled with a global tuning factor of $10^{-5}$ (Jung et al., 2010; Westervelt et al., 2013). In addition, we use a binary nucleation scheme (sulfuric acid and water) when ammonia mixing ratios are less than 1 pptv (Vehkamaki et al., 2002). Secondary organic aerosol (SOA) sources include a biogenic monoterpene component of 19 Tg yr$^{-1}$ as well as an

anthropogenically enhanced component of 100 Tg yr$^{-1}$ (we do not know if the anthropogenically enhanced SOA is due to anthropogenic VOCs or an enhancement of biogenic SOA due to anthropogenic influences on chemistry) (D' Andrea et al., 2013; Spracklen et al., 2011). SOA is condensed irreversibly onto existing aerosol at the time of emission of the parent compounds using 10% of monoterpene emissions for the biogenic component and 0.2 Tg-SOA per Tg-CO for the anthropogenic component (CO emissions are used as a proxy for the spatial and temporal distribution of anthropogenically

enhanced SOA).

In all simulations we use the following emission inputs. We use fossil-fuel combustion anthropogenic emissions from the Emissions Database for Global Atmospheric Research (EDGAR; Janssens-Maenhout et al. 2010) version 4. EDGAR emissions are overwritten in the United States by the Environmental Protection Agency 2011 National Emissions Inventory (NEI2011; http://www3.epa.gov/ttn/chief/), in Canada by Criteria Air Contaminants (CAC;

http://www.ec.gc.ca/inrp-npri/), in Mexico by Big Bend Regional Aerosol and Visibility Study (BRAVO; Kuhns et al. 2005), in Europe by the Cooperative Programme for Monitoring and Evaluation of the Long-Range Transmission of Air Pollutants in Europe (EMEP; Vestreng et al. 2009), and in Asia by the Streets inventory (Streets et al. 2003). Black and organic carbon (OC) emissions from biofuel and fossil-fuel combustion are from Bond et al. (2007) and biomass burning from Global Fire Emissions Database version 3 (GFED3; van der Werf et al. 2010) with three-hourly scale factors.

To estimate the impact of waste-combustion emissions, we include the Wiedinmyer et al. (2014) inventory to the above emissions set-up. Table 1 describes all simulations used in this study. Our BASE simulation uses total BC, OC, SO$_2$, and NH$_3$ mass as in Wiedinmyer et al. (2014). We emit 1% of SO$_2$ as aerosol-phase sulfate (SO$_4$) and use a fixed OA to OC ratio of 1.8. We scale bulk BC and OA mass into size sections assuming a single lognormal mode with a geometric number-mean diameter (GMD) of 100 nm and a standard deviation of 2. We note that Wiedinmyer et al. (2014) does not include

information on the emission size distribution, and this is a significant source of uncertainty (L. Lee et al., 2013). The spatial distribution of emissions is shown in Figure 1 for OA, BC, and NH$_3$. The emissions of SO$_2$ show a similar spatial pattern, but with a factor of 2 lower in magnitude than NH$_3$. The regions with the largest particle emissions are eastern China, northern India, eastern Europe, and Mexico. To test the sensitivity of aerosol radiative effects to waste-combustion emission size distribution, we include two sensitivity simulations, SIZE200 and SIZE30. Here, we conserve total emission mass, but

increase the GMD from 100 nm to 200 nm and decrease the GMD from 100 nm to 30 nm, respectively. These distributions are chosen to reflect the range of reported values from fresh and aged fossil fuel and biomass burning (Ban-Weiss et al., 2010; Carrico et al., 2010; Janhäll et al., 2004; Sakamoto et al., 2015). In Kodros et al. (2015), we also tested the sensitivity of aerosol radiative effects to uncertainty in the standard deviation of the emission size distribution and find the results are qualitatively similar to uncertainties in the GMD (a narrowing of the emission size distribution has a similar effect as to





making the GMD smaller, and visa versa), and so we do not include that here. In PRIMARY, we emit only aerosol-phase BC and OA waste combustion, and remove emissions of gas-phase $SO_2$ and $NH_3$, from this source. This simulation will test the contribution of nucleation and condensation from gas-phase species emitted from waste combustion. Finally, we test the sensitivity of aerosol effects to a factor of 2 uncertainty in total waste-combustion emission mass (HIGHMASS and

LOWMASS). The factor of 2 uncertainty is similar to uncertainties listed in emission factors in Akagi et al. (2011). We compare these simulations to a control simulation with no waste-combustion emissions (WASTE_OFF).

### 2.2 Aerosol optical depth comparison

We compare model aerosol optical depth (AOD) with measurements made with the Aerosol Robotic Network (AERONET). AERONET is a network of ground-based remote sensing instruments aimed at providing global, long-term measurements of

aerosol properties. AERONET uses a sun photometer to measure AOD at several wavelengths in 15 minute intervals when the sun is not obscured by clouds (Holben et al., 1998).

   To compare to AERONET measurements, we output GEOS-Chem-TOMAS size-resolved mass and number concentrations at 3 hour intervals for the BASE and WASTE_OFF simulations. We calculate model AOD using refractive indices from the Global Aerosol Database (Koepke et al., 1997) and published Mie code (Bohren and Huffman, 1983). In

this calculation, we only consider an external mixing-state assumption, though we note the extinction optical depth (what we are referring to as AOD) of an internal mixture is very similar (though the individual scattering and absorption optical depths are different). We average AERONET AOD into three-hour intervals and match measured time and locations with model AOD. We restrict observed AOD to contain greater than 6 measurements within each three-hour time period, thus removing intervals where measurements were limited due to clouds or instrument error. We also restrict our comparison to only

include time periods where the observed AOD is greater than 0.01 to avoid detection-limit issues, as this is the uncertainty limit of AERONET measurements (Holben et al., 1998). We then compare the time average of the co-sampled sites that match these criteria.

### 2.3 Direct radiative effect calculation

We calculate the all-sky direct radiative effect (DRE) using GEOS-Chem-TOMAS monthly averaged aerosol mass and

number concentrations and refractive indices from GADS. We calculate aerosol optical depth, single scatter albedo, and asymmetry parameter using Mie code published in Bohren and Huffman (1983). We use these aerosol optical properties and monthly mean albedo and cloud fraction from GEOS5 as inputs to the offline version of the Rapid Radiative Transfer Model for GCMs (RRTMG; Iacono et al., 2008), implemented in GEOS-Chem (Heald et al., 2014), that considers longwave and shortwave and treats clouds with the Monte Carlo independent column approximation (McICA; Pincus et al., 2003).

We estimate the DRE for six different optical assumptions: a core-shell morphology with and without absorptive OA, an external mixture with a constant absorption enhancement factor of 1.5 (ext*1.5) with and without absorptive OA, a homogeneous internal mixture, and an external mixture. These optical assumptions are discussed in detail in Kodros et al.



(2015) (see Table 1 in Kodros et al. (2015) and Table 3 in this paper). In all cases, we assume the particles are spherical. Briefly, the homogeneous internal mixture assumes all particles in a size bin have the same composition, and the refractive index of the particle is a volume-weighted average of the individual species. The core-shell mixture assumes black carbon forms a spherical core and the other hydrophilic species form a homogeneously mixed shell around the core. The external

mixture assumes black carbon exists as a separate population from the other homogeneously mixed species. Finally, in ext*1.5 we simply multiply the absorption predicted in the external mixture by a constant enhancement factor of 1.5. This is done in some models to estimate the enhanced absorption observed in core-shell mixtures (such as Wang et al., 2014). In the BrC calculations, we estimate absorptive OA (or brown carbon) using the modeled OA to BC ratio and the parameterization in Saleh et al. (2014). In all other calculations, we assume that OA is near-purely scattering throughout the visible spectrum.

We note several important limitations in our treatment of aerosol optics. First, mixing state may vary regionally and temporally, yet here we apply each mixing state globally and at all times. Second, it has been suggested that the optical properties of OA may change with photolysis, yet here our optical properties are held fixed (Lee et al., 2014; Zhao et al., 2015). Therefore, these mixing states should be considered as ideal cases. Finally, we use monthly mean concentrations and properties as well as monthly mean cloud properties. Despite these limitations, we feel this method is sufficient for a first

estimate of the DRE from waste-combustion emissions.

### 2.4 Cloud-albedo aerosol indirect effect calculation

We use monthly mean GEOS-Chem-TOMAS mass and number concentrations and the activation parameterization of Abdul-Razzak and Ghan (2002) to calculate cloud droplet number concentrations (CDNC). We assume a constant updraft velocity of 0.5 m s$^{-1}$. For the AIE, we assume aerosol species are mixed internally within each size bin, and calculate kappa,

the hygroscopicity parameter, as a volume-weighted average of the individual species (Petters and Kreidenweis, 2007). As a control, we assume an effective cloud drop radii of 10 μm and then perturb this value by taking the ratio of CDNC with and without waste-combustion emissions to the one-third power (as in Rap et al., 2013; Scott et al., 2014). To simulate the change in the top-of-the-atmosphere radiative flux due to changes in effective cloud drop radii, we use RRTMG with monthly mean cloud fraction, temperature, pressure, and liquid water content from GEOS5. Again, we note that the use of

monthly mean aerosol and meteorology fields is a limitation in this method, yet we feel it is sufficient for a first estimate.

### 3 Model Results

### 3.1 Comparison to observations

Figure 2 shows comparisons of time-averaged model AOD with AERONET measured AOD at all available sites for the BASE (a-c) and WASTE_OFF (d-f) simulations at 380 nm, 500 nm, and 1020 nm wavelengths (one dot for each site). After

we apply the criteria described in Section 2.2, we have 173 sites at 380 nm, 193 sites at 500 nm, and 211 sites at 1020 nm. The model is generally capable of capturing the trend in measured AOD (in WASTE_OFF at 500 nm slope=0.94 and log





mean bias=-0.18). Given the coarse spatial resolution of the model, the large variability is expected ($r^2$=0.59), as more- and less-polluted areas may exist within the same model gridbox (~400 km length scale). Including waste-combustion emissions slightly improves the model comparison in all three metrics and at all wavelengths (with the exception of the slope at 1020 nm). While this model improvement is minor, it shows that the inclusion of waste-combustion $PM_{2.5}$ does move the model in

the right direction relative to AOD measurements.

### 3.2 Changes in mass concentrations from domestic-waste combustion

Figure 3 shows the annual-mean percent change in boundary-layer OA, BC, $NH_3$, and $PM_{2.5}$ in the BASE-WASTE_OFF simulations, and global-mean OA and BC changes are in Table 2. Combustion of domestic waste under the BASE emission assumptions increases BC and OA concentrations in the global mean by 8.5% and 5.6%, respectively. Regionally, domestic-

waste combustion increases surface BC concentrations by 10-20% throughout much of Asia, northern Africa, and South and Central America. The largest increases in BC (greater than 40%) occur in central Mexico (i.e Mexico City), Turkey, northern Egypt (i.e. Cairo), Sri Lanka, and Papua New Guinea, as well as increases of 30-40% in Pakistan, Morocco, and Central America. Despite greater OA emissions, the relative change of OA from domestic-waste combustion is generally less than BC due to large OA sources from biofuel combustion and secondary organic aerosol in waste-combustion regions. Waste

combustion does increase OA concentrations by more than 40% in Morocco, Turkey, Egypt and the eastern portions of the Middle East. Increases in BC and OA lead to increases in surface $PM_{2.5}$ concentrations of 1-5% over most land masses outside of North America and Australia, and 5-10% in source regions.

Figure 3 also shows the percent change in gas-phase $NH_3$. Waste combustion contributes increases in $NH_3$ of 1-10% in Mexico, Central and South America, and Africa. These are regions of significantly lower $NH_3$ concentrations compared to

Asia and Europe in our simulations, which have negligible relative changes to $NH_3$. In our model, $NH_3$ may contribute to particle nucleation and growth, which may alter particle size distributions (but have little effect on $PM_{2.5}$ mass). These changes are discussed in the next section. Relative changes in $SO_4$ mass generally follow the spatial pattern of OA; however, these changes are less than 1%, and so are not shown.

Table 2 shows the global, annual-mean percent change in boundary-layer BC and OA mass for BASE and the

sensitivity simulations compared to WASTE_OFF (thus isolating the contribution of domestic-waste combustion to BC and OA). The largest changes to modeled BC and OA mass come when emission BC and OA mass is doubled (HIGHMASS) and halved (LOWMASS). In these simulations, the model response to changes in emission mass is close to linear. In simulations, PRIMARY, SIZE30, and SIZE200 total BC and OA emission mass is held constant. The BC and OA mass distributions in SIZE30 and SIZE200 are shifted to smaller and larger sizes, respectively, which may lead to changes in

deposition rates; however, these changes prove to be minor (Table 2). The global-mean relative change in BC and OA mass from domestic-waste combustion ranges from 4.23-16.9% and 2.80-11.2% given the factor of 4 uncertainty in emission mass tested here.



### 3.3 Changes to size-resolved number concentrations

Modeled percent changes to annual-mean size-resolved number concentrations for the BASE-WASTE_OFF comparison are shown in Figure 4 and the global averages in Table 2. Here we consider the following size classes: all particles with a diameter greater than 10 nm (N10), 40 nm (N40), 80 nm (N80), and 150 nm (N150). N40, N80, and N150 are proxies for

climate-relevant particles. In the global mean for our BASE assumptions, waste combustion leads to a larger increase in N80 (1.04%) and N150 (1.16%) than in N10 (0.86%) and N40 (0.90%). This is due to particle emissions represented as a single lognormal mode centered around 100 nm as well as a microphysical feedback (discussed below).

Regionally, the sign and magnitude of particle number varies with size class. Waste combustion tends to increase particle number in each size class in Mexico, Central and South America, and the Pacific Islands. Conversely, waste

combustion decreases N10 and N40 in much of Europe and Asia. This discrepancy is caused by a feedback in aerosol microphysics (discussed in detail in Westervelt et al. 2014, Pierce and Adams et al. 2009, Kodros et al. 2015). Briefly, emitted primary accumulation-mode particles (in BASE this is centered around 100 nm) increases particle surface area, thus increasing the sink for condensation vapors (e.g $H_2SO_4$, $NH_3$, organics). This increased condensation sink reduces nucleation rates, reduces growth rates, and increases coagulational scavenging of nucleation mode particles. This effect is predominant

in Europe and Asia and for smaller size cutoffs (N10 and N40) with a significant contribution of nucleation mode particles (such that waste combustion emissions cannot compensate for the reduced nucleation and growth). The regions of Central and South America, and Africa have lower concentrations of condensable vapor than Europe and Asia in WASTE_OFF, and so this feedback is less dominate. In addition, the Americas and Africa have a larger fractional increase in $NH_3$ concentrations, which aid in nucleation and growth (Figure 3).

Figure 5a shows the change in global-mean particle number distribution for each of the sensitivity simulations relative to WASTE_OFF. In the BASE-WASTE_OFF comparison, particle number decreases for sizes less than 80 nm (due to the feedback described above), and increases for sizes greater than 80 nm (due to primary emissions) (see Table 2 for integrated values). The emission mass scale factor tends to scale the total waste combustion number distribution up or down, while maintaining the general shape of the distribution. The HIGHMASS simulation has a larger increase in accumulation-

mode particles as well as a stronger suppression of nucleation/growth and thus a larger decrease in nucleation-mode particles relative to BASE. Meanwhile, LOWMASS scales in the opposite direction. In the PRIMARY simulation, we remove gas-phase $NH_3$ and $SO_2$ emissions from waste combustion, which contributes to particle nucleation and condensational growth. The removal of these condensable vapors results in a larger suppression of nucleation and growth relative to BASE, causing a larger decrease in sub-80nm particles as well as fewer particles in the 80-200 nm range. When only primary BC and OA

from waste combustion are emitted, the global-mean N10 decreases (-0.17%, Table 2). Nucleation and condensational growth is a less significant contributor to particles greater than 200 nm, and so PRIMARY and BASE have the same distribution at these larger sizes.



Altering the GMD of the emissions size distribution leads to the largest changes in modeled distributions. In SIZE200, we increase emission GMD while keeping total mass constant. Assuming a constant density, this necessitates a smaller number emitted particles (as larger particles have more mass). The reduced particle number relative to BASE allows for nucleation through emitted $NH_3$ and $SO_2$ to increase the number of sub-80 nm particles; however, increases in global-

mean particle number are less than 1% in all size classes (Table 2). Conversely, global-mean particle number increases by greater than 4% in all size classes in the SIZE30 simulation. The increased particle number does lead to a stronger suppression of nucleation; however, as emission GMD is now 30 nm, particle number changes become positive at 20 nm (compared to 80 nm for BASE). Across our simulations, our particle number is most sensitive to making our assumed size distribution smaller.  Thus, understanding near-source, sub-grid-scale aging, which takes primary combustion particles from

diameters around 30 nm to larger sizes (e.g. Sakamoto et al. 2016), is very important in determining the size-distribution effects of waste combustion. Across all sensitivity simulations, we find a range in the global-mean particle number changes due to waste combustion of -0.2 to 4.0% for N10, 0.2 to 1.8% in N40, 0.5 to 5.4% in N80, and 0.4 to 4.4% in N150.

**3.4 Aerosol direct radiative effect**

Table 3 and Figure 6 shows the global-mean all-sky DRE from waste combustion for the six different mixing-state

assumptions for the BASE-WASTE_OFF comparison as well as other sensitivity simulations. Under the BASE assumptions, the global-mean DRE is negative for all mixing states, ranging from -5 mW m$^{-2}$ to -20 mW m$^{-2}$. Globally, the OA to BC ratio from waste-combustion emissions is around 14 to 1, leading to increased scattering and a predominantly negative or cooling DRE. The relative emission mass of OA and BC is a source of uncertainty in Wiedinmyer et al. (2014). In Kodros et al. (2015), we found that the sign of the DRE is sensitive to the OA to BC ratio, where increasing the relative amount of BC

leads to more positive (less negative) DRE. As in previous studies, the DRE becomes more positive when BC is assumed to be mixed internally (represented as core-shell, homogenous, and ext*1.5) as opposed to externally mixed and when OA is assumed to absorb as brown carbon.

Figure 7 shows maps of annual-mean all-sky DRE from waste combustion for all mixing states for BASE-WASTE_OFF. Waste combustion causes a largely negative DRE in much of Asia, Eastern Europe, and the Americas. In

regions with bright surfaces (high albedo), such as the Sahara desert or Greenland ice sheet, the aerosol mixture has a high-enough absorbing component to decrease planetary albedo and cause a positive or warming DRE. The largest-magnitude DRE from waste combustion occurs in eastern China (ranging from -0.4 to -0.1 W m$^{-2}$), as well as northern India and Eastern Europe (-0.3 to -0.05 W m$^{-2}$).

Across all simulations and mixing-state combinations, the global-mean DRE from waste combustion ranges from -

40 mW m$^{-2}$ to +4 mW m$^{-2}$ (Table 3). In simulations HIGHMASS and LOWMASS, the factor-of-2 uncertainty in emission mass leads to approximately a factor-of-2 scale in DRE. In PRIMARY, the removal of the emission gas-phase condensable vapors results in a reduction of sub-200 nm particles, which does not greatly affect the volume distribution (Figure 5b). As a result, the global-mean DRE in PRIMARY is similar to that in BASE. The slightly more negative DRE in the internal





mixtures (core-shell and homogenous) may be a result of smaller shell diameters due to reduced condensation. Both SIZE200 and SIZE30 result in less negative DRE. In SIZE200, the mass distribution is moved to larger sizes resulting in decreases in mass scattering and absorption efficiencies (the size distribution in the BASE simulation has nearly optimal mass scattering and absorption efficiencies). However, the fractional decrease in scattering efficiency is greater than that for

absorption, resulting in a lower single-scattering albedo and an overall decrease in the magnitude of cooling. In SIZE30, the mass distribution is shifted to smaller sizes, which again lowers mass scattering and absorption efficiencies with a greater decrease in scattering relative to absorption. Again, this lowers the single-scattering albedo and decreases the magnitude of cooling.

### 3.5 Cloud-albedo aerosol indirect effect

Figure 8 shows the cloud-albedo AIE from domestic-waste combustion in the BASE-WASTE_OFF simulation. Waste-combustion emissions increase CDNC, and thus cloud reflectivity leading to a negative AIE in Central and South America, Africa, and the Pacific Islands. These are regions where the microphysical feedback suppressing nucleation and growth is less dominate, and waste combustion leads to increases in N40 (Figure 4). Conversely, despite increases in aerosol mass from waste combustion in Asia and Europe, the AIE is zero or slightly positive. This is caused by feedbacks in aerosol and

cloud microphysics. The 'suppression of nucleation and growth' feedback (described earlier) leads to decreases in N40 and thus CDNC. In addition, heavy anthropogenic pollution in India and China results in a strong competition for water vapor. Increases in N80 and N150 (Figure 4) decrease the maximum supersaturation achieved in updrafts, which further limits droplet activation. As a result of the competing regional effects, the global-mean AIE for BASE-WASTE_OFF is -13 mW m$^{-2}$ (Table 3).

20           The global, annual-mean AIE for all simulations is listed in Table 3 and plotted in Figure 6. In the global mean, the AIE ranges from -4 mW m$^{-2}$ to -49 mW m$^{-2}$ based on all emission uncertainties. In general, AIE is stronger in simulations with greater particle-number increases from waste combustion (SIZE30, HIGHMASS) and weakest in simulations with smaller particle-number increases (SIZE200, PRIMARY, LOWMASS). The largest-magnitude AIE occurs in the SIZE30 simulation (-49 mW m$^{-2}$) due to the large increase in particle number (Table 2 and Figure 5). The lowest magnitude AIE

occurs in the PRIMARY simulation, where condensable vapors from waste-combustion emissions are removed, leading to fewer particles in the 40-200 nm size range relative to BASE (Figure 5). This implies a non-trivial contribution of nucleation and growth to CCN concentrations from the NH$_3$ and SO$_2$ emissions from trash combustion.

## 4. Conclusions

In this paper, we use a global chemical-transport model to provide a first estimate of aerosol concentrations and radiative

impacts from open, uncontrolled combustion of domestic waste. Using emissions from Wiedinmyer et al. (2014), we find increases in global-mean boundary-layer BC and OA mass of 8.5% and 5.6%, with regional increases exceeding 40%,





relative to simulations without waste-combustion emissions. In our base simulations, we estimate domestic-waste combustion emissions lead to a global-mean DRE -5 to - 20 mW m$^{-2}$ (range due to optical assumptions) and an AIE of -13 mW m$^{-2}$. In areas where open waste-combustion occurs, the resulting aerosol radiative impacts can range from -0.1 to -0.4 W m$^{-2}$. On a global scale, we find minor improvements of model AOD compared to observed AERONET AOD when including

waste-combustion emissions.

We include several sensitivity simulations to explore uncertainties in waste-combustion emission size distribution, gas-phase species (leading to secondary aerosol), and emitted particulate mass. In the global, annual mean, we estimate a range in the DRE of +4 to -40 mW m$^{-2}$ and in the AIE of -4 to -49 mW m$^{-2}$. We find altering the waste-combustion emission size distribution while keeping total mass constant (SIZE200 and SIZE30) leads to substantial changes in the modeled size

distribution, thus affecting the AIE (-8 to -49 mW m$^{-2}$) and to a lesser extent the DRE (+4 to -18 mW m$^{-2}$). In PRIMARY, we emit only primary BC and OA particles from waste combustion (and not particle-precursor gases). In this simulation, waste combustion leads to a stronger suppression of nucleation and growth, smaller increases in particle number, and a lower magnitude AIE (-4 mW m$^{-2}$). Thus, the particle-precursor gases emitted from waste-combustion are important contributors to new-particle formation and growth, and they contribute to the negative AIE from waste-combustion emissions. We introduce

a factor of 2 uncertainty in emission mass in HIGHMASS and LOWMASS. We find this leads to a near-linear increase in BC and OA mass and DRE (-2.5 mW m$^{-2}$ to -40 mW m$^{-2}$). Scaling emission mass up/down also increases/decreases emitted particle number, leading to a range of AIE of -7 to -24 mW m$^{-2}$.

Our estimates for the aerosol radiative effects from waste combustion are relatively small compared to the total aerosol burden; however, they are of similar magnitude to recent estimates of radiative impacts from biofuel combustion (-

0.06 to +0.08 W m$^{-2}$ for the DRE and -0.05 to +0.01 W m$^{-2}$ for the AIE), another emission source associated with developing countries (Bond et al., 2013; Butt et al., 2016; Kodros et al., 2015; Lacey et al., 2015). Based on our estimates, it appears that the aerosol radiative effects from waste combustion likely have an overall cooling effect.

In a separate study (Kodros et al., 2016), we estimated that this waste-combustion aerosol leads to 284,000 (95% CI: 188,000-404,000) premature deaths globally each year. If waste-combustion aerosol is reduced to protect human health,

this may also lead to positive radiative forcing (warming tendency) due to a reduction of the waste-combustion-aerosol cooling. A reduction in waste combustion would also lead to a change in greenhouse gases, although these greenhouse-gas estimates are out of the scope of this paper due to the complex nature of greenhouse-gas emissions from waste decomposition and/or recycling. The greenhouse-gas changes due to choice of waste-removal method is an important topic to be studied in the future.






## Acknowledgements

This research has been supported by a grant from the U.S. Environmental Protection Agency's Science to Achieve Results (STAR) program through grant #83543801. Although the research described in the article has been funded by the U.S. Environmental Protection Agency's STAR program, it has not been subjected to any EPA review and therefore does not

necessarily reflect the views of the Agency, and no official endorsement should be inferred. David Ridley was funded through National Aeronautics and Space Administration grant: NASA NN14AP38G.

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



Table 1. Description of simulations

| Simulation | GMD[a] | Mass Scale Factor | Species |
|---|---|---|---|
| BASE | 100 nm | 1.0 | primary and secondary |
| SIZE200 | 200 nm | 1.0 | primary and secondary |
| SIZE30 | 30 nm | 1.0 | primary and secondary |
| PRIMARY | 100 nm | 1.0 | primary BC and OA only |
| HIGHMASS | 100 nm | 2.0 | primary and secondary |
| LOWMASS | 100 nm | 0.5 | primary and secondary |
| WASTE_OFF | N/A | 0.0 | None |

[a]GMD = geometric number-mean diameter



Table 2. Percent change in global, annual-mean N10, N40, N80, N150, BC mass, OA mass from domestic-waste combustion (i.e. relative to WASTE_OFF simulation).

| Simulation | N10 [%] | N40 [%] | N80 [%] | N150 [%] | BC [%] | OA [%] |
|---|---|---|---|---|---|---|
| BASE | 0.86 | 0.90 | 1.04 | 1.16 | 8.46 | 5.60 |
| SIZE200 | 0.86 | 0.55 | 0.46 | 0.36 | 8.36 | 5.53 |
| SIZE30 | 4.02 | 5.43 | 5.42 | 4.40 | 8.07 | 5.14 |
| PRIMARY | -0.17 | 0.232 | 0.52 | 0.89 | 8.47 | 5.61 |
| HIGHMASS | 1.68 | 1.77 | 2.06 | 2.31 | 16.9 | 11.2 |
| LOWMASS | 0.44 | 0.45 | 0.53 | 0.59 | 4.23 | 2.80 |



Table 3. Global-mean all-sky direct radiative effect and cloud-albedo aerosol indirect effect relative to the WASTE_OFF simulation for all sensitivity simulations and mixing-state assumptions

| Simulation | Direct Radiative Effect [mW m$^{-2}$] | | | | | | AIE [mW m$^{-2}$] |
|---|---|---|---|---|---|---|---|
| | Core-Shell, BrC | Ext*1.5, BrC | Homogenous | Core-Shell | Ext*1.5 | External | |
| BASE | -5.0 | -8.7 | -11 | -13 | -17 | -20 | -13 |
| SIZE200 | +3.2 | -6.0 | -3.7 | -8.7 | -16 | -18 | -8.3 |
| SIZE30 | +4.1 | +3.7 | -1.5 | -1.6 | -2.3 | -5.8 | -49 |
| PRIMARY | -5.7 | -9.3 | -12 | -14 | -17 | -20 | -4.0 |
| HIGHMASS | -10 | -17 | -22 | -26 | -34 | -40 | -24 |
| LOWMASS | -2.5 | -4.3 | -5.6 | -6.5 | -8.7 | -10 | -6.7 |




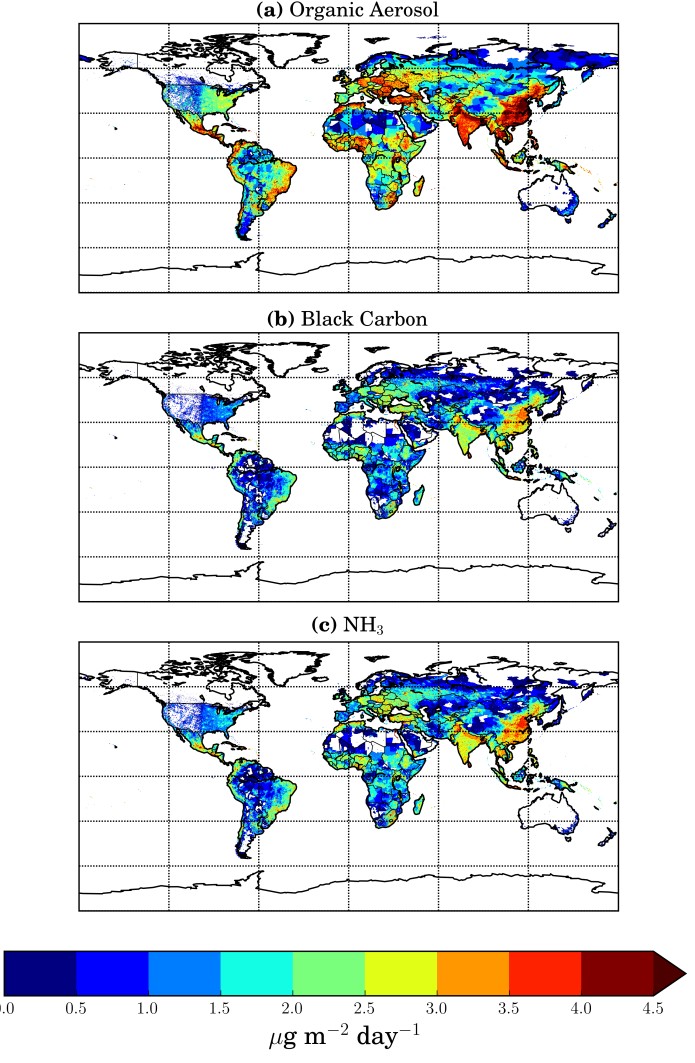

Figure 1. Emission fluxes from the open combustion of domestic waste for (a) organic aerosol, (b) black carbon, and (c) NH$_3$ from Wiedinmyer et al. (2014).





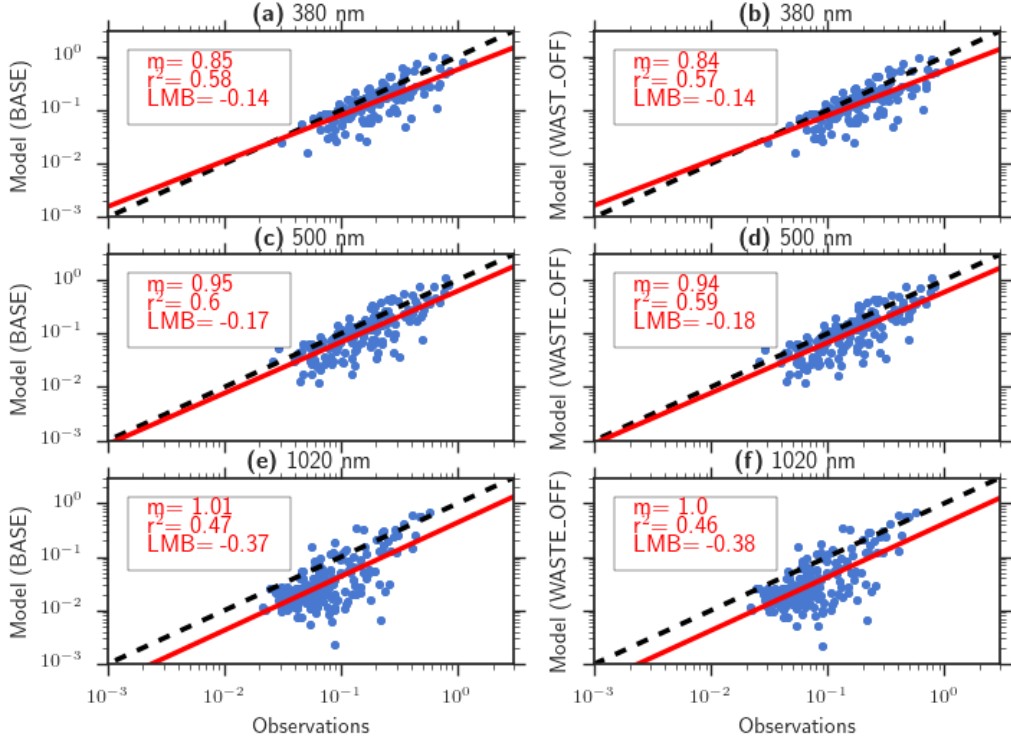

Figure 2. One-to-one plots of time-average AOD for BASE (a,c,e) and WASTE_OFF (b,d,f) compared to AERONET observations at 380 nm (a,b), 500 nm (c,d), and 1020 nm (e,f). The values in the box give the slope of the linear regression (m), the correlation ($r^2$), and the log-mean bias (LMB), while the black dashed line represents the 1:1 line and the red line is the result of the linear regression.



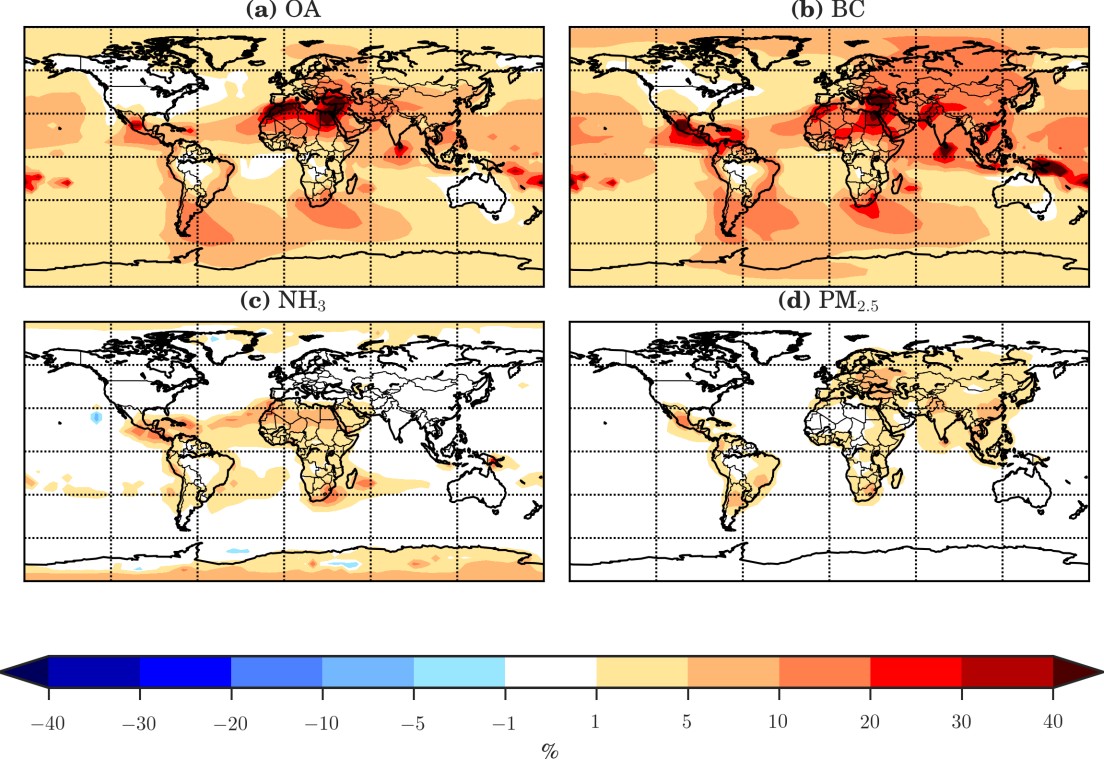

Figure 3. Percent changes in annually averaged boundary-layer (a) OA, (b) BC, (c) $NH_3$, and (d) $PM_{2.5}$ mass due to domestic-waste combustion (BASE-WASTE_OFF).



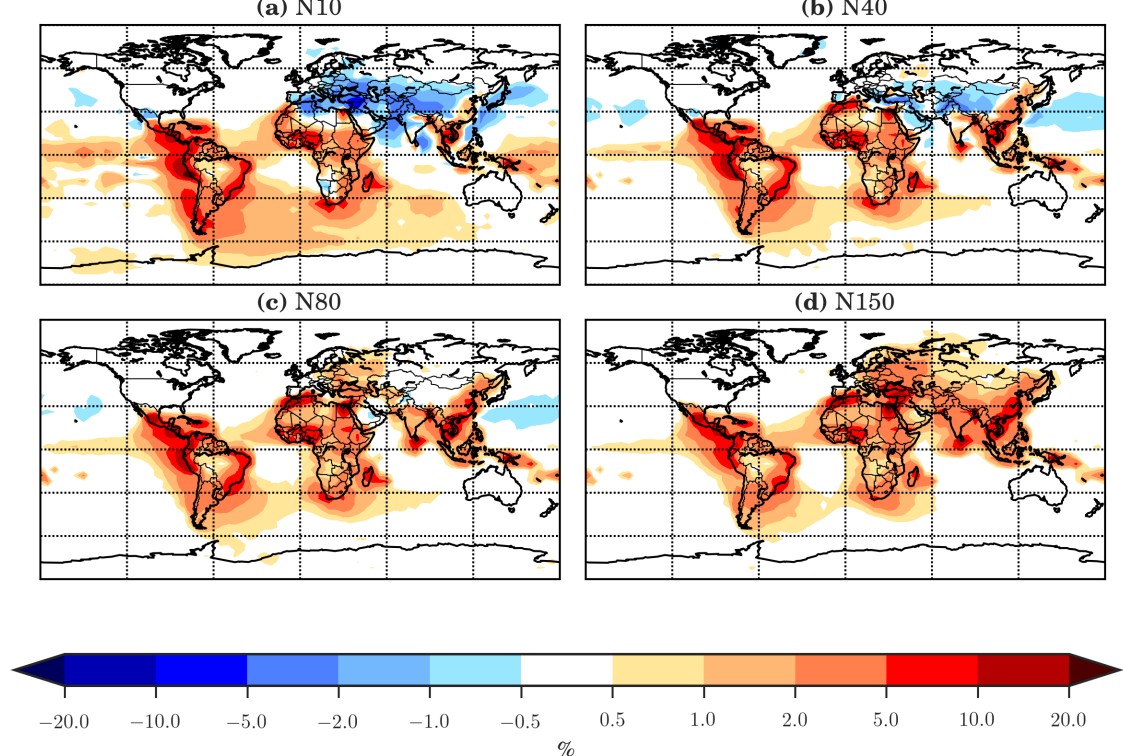

Figure 4. Annual-mean percent change in boundary-layer (a) N10, (b) N40, (c) N80, (d) N150 for domestic-waste combustion (BASE-WASTE_OFF).



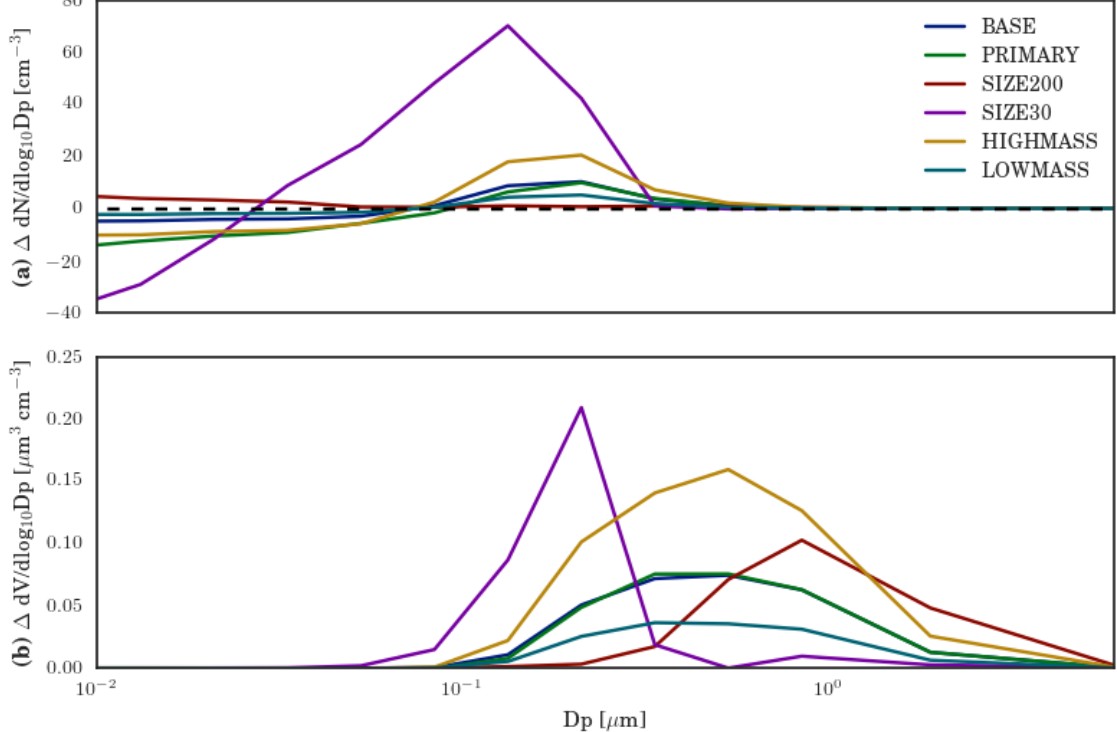

Figure 5. The change in global-mean (a) number distribution and (b) volume distribution for the different sensitivity studies relative to the WASTE-OFF simulation.





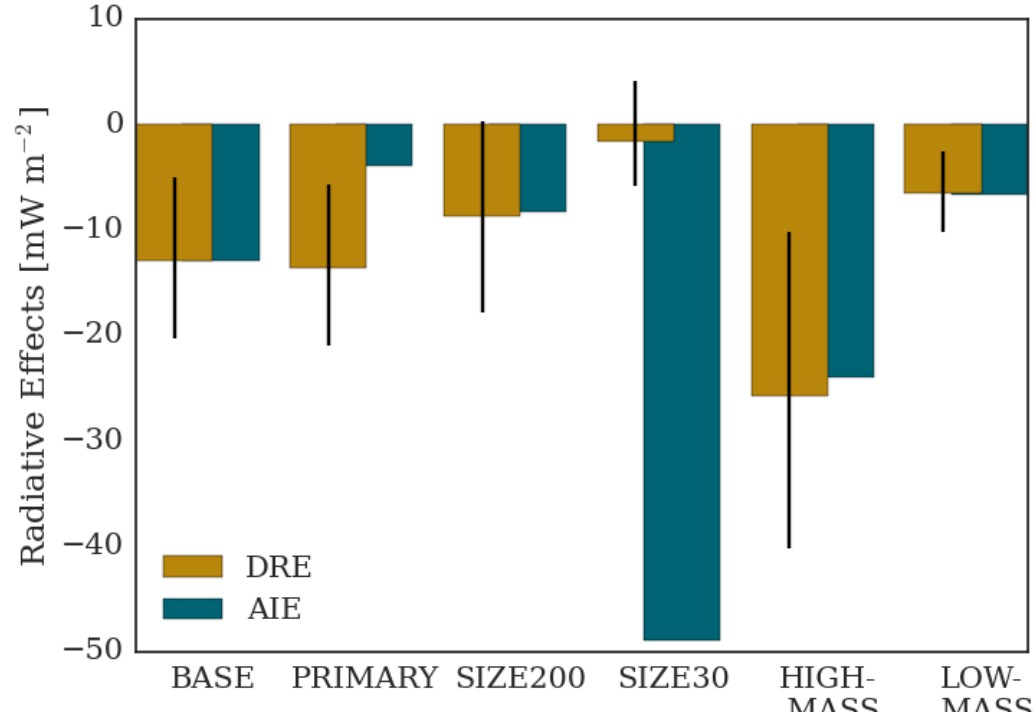

Figure 6. Global-mean all-sky direct radiative effect (DRE) and cloud-albedo aerosol indirect effect (AIE) for the baseline (BASE), aerosol-phase emissions only (PRIMARY), larger (SIZE200) and smaller (SIZE30) geometric mean diameter, and doubled (HIGHMASS) and halved (LOWMASS) emission mass relative to the WASTE_OFF simulation. The gold bars represent the DRE for the core-shell mixing state, while the error bars show the range due to the other optical assumptions.





Figure 7. The all-sky direct radiative effect for domestic-waste combustion (BASE-WASTE_OFF) assuming a (a) core-shell with brown carbon, (b) homogenous internal, (c) core-shell, (d) external with brown carbon and an absorption enhancement factor of 1.5, (e) external with an absorption enhancement factor of 1.5, and (f) external mixing state.



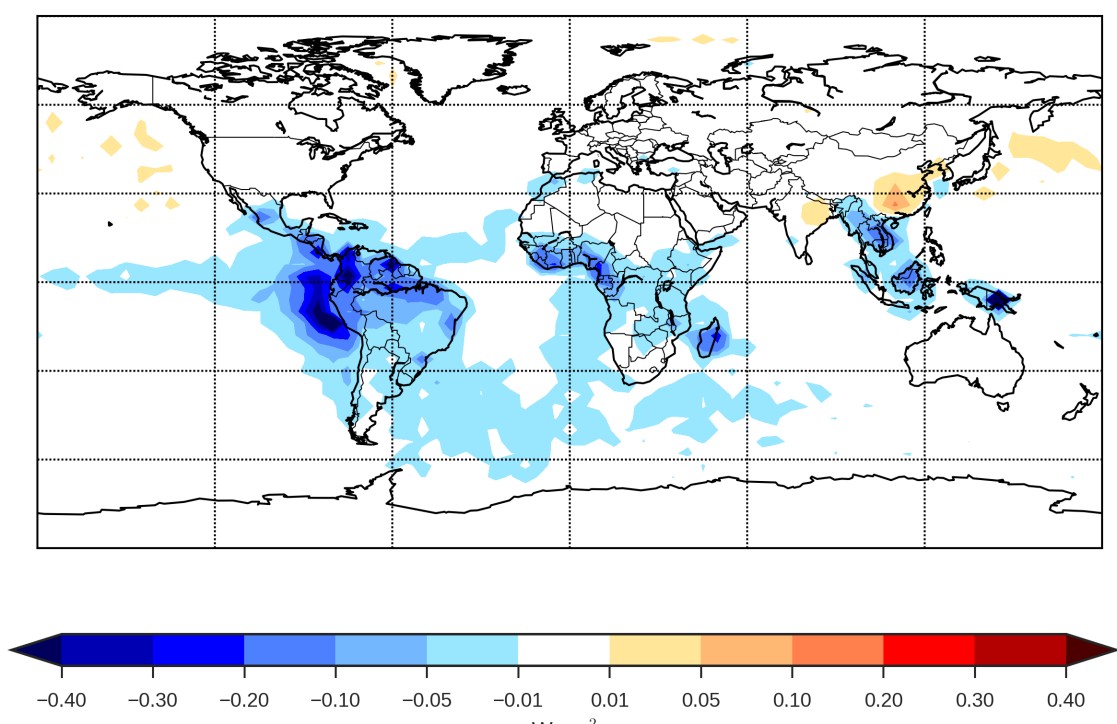

Figure 8. The cloud-albedo aerosol indirect effect difference between BASE-WASTE_OFF.