# Peer review of "The aerosol radiative effects of uncontrolled combustion of domestic waste"

_Atmospheric Chemistry and Physics, 2016_

## Referee Comment (RC1) · Anonymous Referee #1 · 9 May 2016

The paper makes a first estimate of the impacts of uncontrolled domestic waste combustion on atmospheric aerosol and the impacts on climate. This emission source is very rarely included in atmospheric aerosol models and this study is an important first step towards understanding the impacts on aerosol and climate. The manuscript is very well written and I suggest publication after the following very minor comments have been accounted for.

Minor comments

Page 3, Line 27. These simulations have a very coarse spatial resolution (4x5o). The authors recognise this weakness.

Page 6, Section 3.1 The model evaluation using AERONET AOD demonstrates that including the waste combustion emission source does not degrade the model. The

authors could consider using measured BC or OC mass concentrations from regions heavily impacted by waste combustion as an additional evaluation of the model.

Page 6, line 31: Suggest reword to remove the word "trend".

---

## Referee Comment (RC2) · Anonymous Referee #2 · 11 May 2016

This paper presents an assessment of the direct and indirect radiative forcing due to aerosol particles that are emitted by the uncontrolled combustion of domestic waste. The emission inventory of waste-combustion emissions has large uncertainties. The authors address this issue by performing a suite of sensitivity simulations that vary the size parameter or the emission strength of the waste-combustion emissions, as well as the assumptions needed for calculating aerosol optical properties. The paper is interesting since this emission source is not generally included in global emission inventories but (as the authors show) contributes to the absorbing aerosol burden in certain regions of the globe. This study is one of the first that attempts to quantify the climate impact of this aerosol type and the topic fits well into the scope of this journal. I consider the paper worthy of publication in ACP after the following comments are addressed.

[Figure]

General comment on results: As the authors point out, many assumptions about the domestic waste emissions have large uncertainties. Since this is one of the first studies that investigates climate impacts of waste-combustion emissions, it would be useful to formulate explicitly in the conclusions, which aspects of the emission information would be most useful to improve in future work.

General comment on presentation: Some phrases are rather colloquial, and I suggest going through the paper and rephrase them. Examples are:

- p. 3, line 16: look at -> quantify.

- P. 5, line 1: making the GMD smaller -> decrease the value for the GMD

- P. 5, line 19: 6 -> six (write out numbers smaller than ten)

- P. 9, line 3: smaller number emitted particles -> smaller number flux of emitted particles

- The phrase "particle number" is often used when it should be "particle number concentration".

- P. 9, line 8: making our assumed size distribution smaller -> decrease the value for the GMD of the assumed size distribution.

p. 2, line 7: "... how the particles are mixed together": do you mean mixed together within the population or mixed together within one particle?

p.2, second paragraph: Scattering and absorption in general also depends strongly on the morphology of the particles. Assuming different variants of spherical particles may not reflect reality very well.

p. 2, line 25: define kappa.

p. 3-4: What about sea salt and dust emissions?

p. 3-4: Does the chemistry model include ammonium nitrate? If not, can you say

something how this might impact the conclusions? p. 5: Direct radiative effect calculation: are the six different optical assumptions applied to all BC containing particles, regardless of their source (i.e. not only to the waste combustion particles)?

p. 9, line 29/30: the minus sign was separated from the number due to a line break. Make sure that this doesn't happen.

p. 8, line 18: typo: dominant

Figure 2: To draw the conclusion that the BASE run and the WASTE_OFF run are different, you need to show the differences in slope and r2 are statistically significant. If it turns out that they are indeed different, please comment if the effect size is meaningful (i.e. of practical importance).

―――――――――――――――――――――

---

## Author Comment (AC1) · 16 May 2016

We thank Reviewer 1 for their constructive suggestions. We reproduce the comments here in italics.

*Page 3, Line 27. These simulations have a very coarse spatial resolution (4x5o). The authors recognise this weakness*

Due to computational limits, we use a coarse spatial resolution. This means we are calculating radiative effects using aerosol concentrations and meteorological variables averaged over quite large (about 400 km) spatial scales. As aerosol and meteorological factors vary on much smaller spatial scales, the coarse spatial resolution likely introduces some errors. This may be more important for the AIE, as cloud reflectance has a nonlinear relationship with CDNC. Despite this, we do not expect these errors to

qualitatively change our results and conclusions.

*The model evaluation using AERONET AOD demonstrates that including the waste combustion emission source does not degrade the model. The authors could consider using measured BC or OC mass concentrations from regions heavily impacted by waste combustion as an additional evaluation of the model.*

This is an interesting suggestion. We did have interest in including measurements of BC/OC mass concentrations; however, robust measurements of mass concentration (especially speciated mass concentrations) are rare in developing countries (which is where most waste combustion emissions are located). One possibility is the SPARTAN network (http://spartan-network.org/), which measures $PM_{2.5}$ concentrations in developing countries. Unfortunately, there are not enough measurements available to achieve robust statistics. We would be interested in hearing about other possible datasets.

*Page 6, line 31: Suggest reword to remove the word "trend".*

We agree. We are replacing the word "trend" with "variability".
* * *

---

## Author Comment (AC2) · 16 May 2016

We thank Reviewer 2 for their constructive response. We reproduce reviewer comments here in italics.

*General comment on results: As the authors point out, many assumptions about the domestic waste emissions have large uncertainties. Since this is one of the first studies that investigates climate impacts of waste-combustion emissions, it would be useful to formulate explicitly in the conclusions, which aspects of the emission information would be most useful to improve in future work*

Thank you for this suggestion and we feel this will be a useful addition to the discussion. We have added the following lines to the conclusions:

"There is little information on emission size distribution and optical properties from waste combustion. Better knowledge of these parameters along with continued validation of emission mass fluxes will reduce model uncertainty."

*General comment on presentation: Some phrases are rather colloquial, and I suggest going through the paper and rephrase them.*

Thank you for suggesting these corrections. We have made a number of edits following the reviewer's examples.

*p. 2, line 7: ". . . how the particles are mixed together": do you mean mixed together within the population or mixed together within one particle?*

We mean mixed together within a single particle. We have rephrased the manuscript to read:

"the manner of mixing of different species within a single particle"

*p.2, second paragraph: Scattering and absorption in general also depends strongly on the morphology of the particles. Assuming different variants of spherical particles may not reflect reality very well.*

This is a good point. We do assume spherical particles (a common assumption in global modeling studies). This will introduce some errors. We have added the following lines:

"In this study, we assume spherical particles, which is not perfectly realistic for fresh combustion aerosol, and these details of particle shape may alter optical properties."

*p. 2, line 25: define kappa.*

We have added the following lines: "Particle hygroscopicity can be described with the hygroscopicity parameter, kappa, which is the ratio of the number of moles of solute per dry volume to the moles of water per volume of pure water (Petters and Kreidenweis, 2007)."

*p. 3-4: What about sea salt and dust emissions?*

Sea salt emissions follow the scheme of Jeagle et al. (2011), and dust emissions follow the DEAD scheme of Zender et al. (2003). We have added this statement and references to the methods section.

*p. 3-4: Does the chemistry model include ammonium nitrate? If not, can you say something how this might impact the conclusions?*

Ammonium nitrate is not included in the TOMAS microphysics; however, it is included in the "bulk" GEOS-Chem aerosol setup, which runs concurrently with TOMAS. We are interested in incorporating ammonium nitrate into TOMAS in the future.

*p. 5: Direct radiative effect calculation: are the six different optical assumptions applied to all BC containing particles, regardless of their source (i.e. not only to the waste combustion particles)?*

This is a good question, and we have added a clarifying sentence to the manuscript. The optical assumptions are applied to all BC containing particles. We do not track particles by emission source after they have been emitted into the model (sources are emitted separately, but once emitted they enter the same tracer). Given the coarse spatial and temporal resolution of our model, it is fair to assume some degree of mixing at the model length scale. In general, BC from different sources (fossil fuel vs. biomass burning) may mix at different timescales due to co-emitted species. Our optical assumptions are therefore idealized cases.

*p. 9, line 29/30: the minus sign was separated from the number due to a line break. Make sure that this doesn't happen.*

In the final typeset version, we will look for this error.

*p. 8, line 18: typo: dominant*

Thank you.

*Figure 2: To draw the conclusion that the BASE run and the WASTEOFF run are different, you need to show the differences in slope and $r^2$ are statistically significant. If it turns out that they are indeed different, please comment if the effect size is meaningful (i.e. of practical importance).*

The differences are not statistically significant. We included this figure first to demonstrate that including this emission source does not degrade the model comparison, and second to show TOMAS has skill at reproducing observed aerosol optical properties (this is the first GEOS-Chem-TOMAS paper that uses GEOS-Chem version 10). We have added the line:

"While these changes are not statistically significant, we note that GEOS-Chem-TOMAS has some skill at reproducing observed aerosol optical properties, and including this inventory does not degrade model comparison"
* * *